# Trends in Congenital Syphilis Incidence and Mortality in Brazil’s Southeast Region: A Time-Series Analysis (2008–2022)

**DOI:** 10.3390/epidemiologia6020022

**Published:** 2025-05-05

**Authors:** Alexandre Castelo Branco Araujo, Orivaldo Florencio de Souza, Betina Bolina Kersanach, Julia Silva Cesar Mozzer, Victor Lopes Feitosa, Vinicius Andreata Brandão, Filomena Euridice Carvalho de Alencar, Norma Suely Oliveira, Andrea Vasconcellos Batista da Silva, Luiz Carlos de Abreu

**Affiliations:** 1Study Design and Scientific Writing Laboratory, Health Sciences Center, Federal University of Espírito Santo, Vitoria CEP29043-900, Brazil; luizcarlos@usp.br; 2Postgraduate Program in Health Sciences, Federal University of Acre, Rio Branco CEP69915-900, Brazil; orivaldo.souza@ufac.br; 3Health Sciences Center, Federal University of Espírito Santo, Vitoria CEP29043-900, Brazil; betina.bk.boker@gmail.com (B.B.K.); juliascmozzer@gmail.com (J.S.C.M.); victorlopesmed@gmail.com (V.L.F.); viniciusandreatab@gmail.com (V.A.B.); filomena.alencar@ufes.br (F.E.C.d.A.); norma.oliveira@ufes.br (N.S.O.); andrea.silva@ufes.br (A.V.B.d.S.); 4Postgraduate Program in Medical Sciences, University of São Paulo, São Paulo CEP01246-903, Brazil

**Keywords:** congenital syphilis, trends, incidence, mortality, Brazil

## Abstract

Congenital syphilis (CS) is an important infectious cause of miscarriage, stillbirth, and neonatal morbidity and mortality. Despite the advances in diagnosis and treatment, CS continues to challenge health systems with increasing incidence and mortality rates in recent years worldwide. Given this, the present study aims to comparatively analyze the temporal trends in CS incidence and mortality in Brazil’s Southeast Region from 2008 to 2022. This is an ecological time-series study using secondary data on congenital syphilis from the states of Espírito Santo, Minas Gerais, Rio de Janeiro, and São Paulo. The data was extracted from the Brazilian Health System Informatics Department. Incidence and mortality rates were calculated per 100,000 live births. Joinpoint regression models were employed to identify trends in annual percentage change and average annual percentage change with 95% confidence intervals. The temporal trend of CS incidence in Brazil’s Southeast Region increased 12.8% between 2008 and 2022. Minas Gerais, São Paulo, Espírito Santo, and Rio de Janeiro showed increasing temporal trends of 21.4%, 14.1%, 14.0%, and 10.9%, respectively. The temporal trend of CS mortality in Brazil’s Southeast Region rose 11.9% between 2008 and 2022. Minas Gerais, São Paulo, and Rio de Janeiro exhibited increasing mortality temporal trends of 21.9%, 20.8%, and 10.1%, respectively. In contrast, Espírito Santo showed reduced mortality, with no deaths in 2021 and 2022. The temporal trend of CS incidence increased in all states of Brazil’s Southeast Region between 2008 and 2022, highlighting the need to reassess control measures. The temporal trend of CS mortality also increased during the same period, except in Espírito Santo. Considering that CS is preventable with adequate prenatal care and low-cost measures, these findings can serve as instruments to support strengthening public health policies.

## 1. Introduction

Congenital syphilis (CS) results primarily from transplacental transmission of *Treponema pallidum* by untreated or inadequately treated pregnant women, with consequent fetal infection [1,2,3]. Despite decades of knowledge about transmission, diagnosis, and treatment, CS continues to challenge global public health, being a substantial cause of miscarriage, stillbirth, neonatal mortality, prematurity, low birth weight, and multisystem congenital infection [3,4,5,6].

In 2016, the World Health Organization (WHO) estimated that there were approximately 661,000 new cases of CS, 143,000 fetal deaths or stillbirths, 61,000 neonatal deaths, and 41,000 premature or low-weight births worldwide [7]. In the same year, the WHO renewed the global initiative of “Elimination of Mother-to-Child Transmission” (EMTCT) of CS to less than 50 cases per 100,000 live births by 2030 in 80% of countries [7,8]. To achieve this goal, at least 95% of pregnant women should have at least one antenatal care visit with syphilis testing, and at least 95% of syphilis-seropositive pregnant women should receive adequate treatment [9,10]. Nevertheless, the WHO Regional Office defined specific initiatives for the European Region, aiming at the target of ≤10 cases per 100,000 live births in 2025 and ≤1 case per 100,000 live births in 2030 [11].

CS is preventable with early diagnosis and appropriate treatment with penicillin during pregnancy [4,7]. However, limitations in prenatal care and testing, inappropriate treatment of the pregnant woman and her partner, and late detection of seroconversion during pregnancy result in missed opportunities for control [2,7,8]. In recent years, there has been a notable resurgence of CS in high-income countries and an increase in incidence in middle- and low-income countries, where poverty, low education, difficulties in accessing healthcare, and risky sexual behaviors contribute to the perpetuation of the problem, generating social and economic impacts [3,9,11].

WHO data indicated a global reduction in CS incidence rates, from 539 to 425 new cases per 100,000 live births between 2012 and 2020, although with regional disparities and higher rates in countries in Africa, America, and the Eastern Mediterranean [4,7,9,11]. A study in the European Union showed a decrease in the incidence of CS, from 3.1 cases per 100,000 live births in 2007 to 1.1 cases in 2017 [12], followed by an increase to 2.7 cases in 2023 [13]. In the United States of America (USA), rates rose from 8.4 cases in 2012 to 77.6 cases per 100,000 live births in 2021 [14,15,16], while in Canada, the increase was from 2.0 cases in 2017 to 13.4 cases per 100,000 live births in 2020 [17].

Temporal analysis in Brazil revealed a rise in CS incidence rates, from 200 to 876 new cases per 100,000 live births between 2007 and 2017 [18]. The temporal trend analysis indicated an annual growth of 26.96% between 2008 and 2013, and 10.25% between 2013 and 2018 [19]. The south, northeast, and southeast regions recorded the highest CS incidence rates in 2018, with 836, 840, and 887 new cases per 100,000 live births, respectively [19].

Brazil is a signatory of the EMTCT initiative. However, in response to the alarming increase in syphilis cases, the Ministry of Health launched the Strategic Actions Agenda for Reducing Syphilis and the “Syphilis No” Project. These initiatives sought to mitigate acquired syphilis, syphilis in pregnant women, and congenital syphilis through collaborative work with governmental and non-governmental organizations and health professional associations. The focus is on improving health practices within the Unified Health System by expanding diagnostic coverage and adequate treatment for pregnant women and their sexual partners during prenatal care, including testing acquisition and treatment supplies distribution in addition to monitoring related actions [8,18,19].

Considering the increasing temporal trend in CS incidence rates in Brazil and in several countries around the world; the need for improvements to reach the WHO target by 2030; the estimates of 68,000 new cases of CS in the Americas in 2022 [20]; the record of 26,468 cases in Brazil, of which 11,581 occurred in the Southeast Region in the same year [21]; and the scarcity of studies on the incidence and mortality from CS in this region, it becomes imperative to update the temporal trends of these epidemiological indicators [18,19,22]. In this context, the present study aims to comparatively analyze the temporal trends of the incidence and mortality from CS in children under 1 year of age in the states of Brazil’s Southeast Region between 2008 and 2022.

## 2. Materials and Methods

### 2.1. Study Design, Location, and Population

This is an epidemiological study, with an ecological design, of the time-series type, based on secondary data from SC in children under one year old in the states of Brazil’s Southeast Region between 2008 and 2022. Information on the number of live births, new cases, and deaths from CS was obtained for Brazil’s Southeast Region. This region is the most populous in the country, with 84,840,113 inhabitants and a population density of 9176 inhabitants/km^2^ [23]. It comprises the states of Espírito Santo, Minas Gerais, Rio de Janeiro, and São Paulo, which are among the ten states with the highest Human Development Indexes in the country [24].

### 2.2. Data Extraction

All information was extracted from the database of the Informatics Department of the Unified Health System (DATASUS) of the Brazilian Ministry of Health (https://datasus.saude.gov.br/informacoes-de-saude-tabnet/ (accessed on 2 October 2024) by place of residence and in each year, from 2008 to 2022. The DATASUS database contains relevant information on the health of the Brazilian population, with public and unrestricted access. Data on live births, new cases, and deaths from congenital syphilis incorporated into the DATASUS database come from the Live Birth Information System, the Notifiable Diseases Information System, and the Mortality Information System, respectively. These national systems receive information continuously from all locations in Brazil and are managed by the Brazilian Ministry of Health. Data on live births were accessed at http://tabnet.datasus.gov.br/cgi/deftohtm.exe?sinasc/cnv/nvsp.def (accessed on 2 October 2024). Information on new cases of SC was accessed at http://tabnet.datasus.gov.br/cgi/deftohtm.exe?sinannet/cnv/sifilisbr.def (accessed on 2 October 2024), while information on deaths was accessed at http://tabnet.datasus.gov.br/cgi/deftohtm.exe?sim/cnv/obt10sp.def (accessed on 2 October 2024).

Data extraction was performed on 16 September 2024 by trained researchers, using the Internet Data Tabulator (TabNet – online program of the Ministry of Health, Brasilia, Brasil) for a file in comma-separated values (CSVs) format. Subsequently, the collected information was transferred to the Microsoft Office Excel spreadsheet. The study variables were extracted by year for Brazil’s Southeast Region and stratified for the states of Espírito Santo, Minas Gerais, Rio de Janeiro, and São Paulo.

### 2.3. Study Variables

The dependent variable was congenital syphilis, following criteria adopted by the Brazilian Ministry of Health: (1) Newborns, stillbirths, or miscarriages of women with syphilis not treated or inadequately treated. (2) Children younger than 13 years old with at least one of the following situations: (a) clinical manifestations, cerebrospinal fluid, or radiological alteration of congenital syphilis and reactive non-treponemal test (Venereal Disease Research Laboratory (VDRL) or Rapid Plasma Reagin (RPR)); (b) non-treponemal test titers of babies higher than their mother’s, in at least two peripheral blood sample dilutions, collected simultaneously at the moment of birth; (c) non-treponemal test ascending titers in at least 2 dilutions in the exposed child’s follow-up; and (d) non-treponemal test titers still reacting after 6 months of age, except in case of therapeutic follow-up. (3) Microbiological evidence of infection by *T. pallidum* in nasal secretion or skin lesion samples; child, miscarriage, or stillbirth biopsy or necropsy [25]. According to the International Classification of Diseases, version 10, congenital syphilis is coded as A50 in the infectious and parasitic diseases chapter [26]. New cases and deaths from congenital syphilis were extracted and analyzed in aggregate, by year.

The incidence of CS per 100,000 live births and mortality per 100,000 live births were calculated per year for the Southeast Region of Brazil and stratified for the states of Espírito Santo, Minas Gerais, Rio de Janeiro, and São Paulo. The incidence of CS was calculated by dividing the number of new cases by the number of live births; the quotient was multiplied by 100,000. Mortality due to CS was calculated by dividing the number of deaths by the number of live births; the quotient was multiplied by 100,000. All calculations were performed using the Microsoft Office Excel spreadsheet.

### 2.4. Statistical Methods and Analysis

The temporal trend of incidence and mortality by SC was estimated by joinpoint regression with the aid of the Joinpoint Regression Program (version 5.2, 2024). The permutation test was applied to select the models. The direction and magnitude of the temporal trends were presented by the Annual Percent Change (APC) and the Average Annual Percent Change (AAPC), with 95% confidence intervals estimated by the parametric method. When there was no trend in the segment during that period, the APC was similar to the AAPC. Joinpoint regression models with a *p*-value equal to or less than 0.05 accepted the hypothesis of occurrence of annual variation. Thus, positive values indicated an increasing trend, and negative values indicated a decreasing trend.

### 2.5. Legal and Ethical Aspects

Due to the study design and considering that all data used are publicly assessable, ethical approval was not required following National Health Council Resolution 510/2016.

## 3. Results

In the comparison between the years 2008 and 2022, the total number of live births in Brazil’s Southeast Region decreased from 1,130,407 to 979,681 (13.3%). There was a reduction in all states, with a drop from 51,852 to 51,729 in the state of Espírito Santo, from 260,916 to 235,063 in the state of Minas Gerais, from 215,844 to 180,369 in the state of Rio de Janeiro, and from 601,795 to 512,520 in the state of São Paulo (Figure 1; Appendix A).

The total absolute number of new cases from CS in Brazil’s Southeast Region in children under one year of age between 2008 and 2022 was 112,481, with an increase from 2392 new cases in 2008 to 10,669 in 2022 (346.0%). The highest values were recorded in 2021 for the state of Rio de Janeiro and 2022 for the state of São Paulo. All states showed an increase in the number of new cases, with an increase from 195 to 2219 in Minas Gerais (1037.9%), from 104 to 562 in Espírito Santo (440.4%), from 836 to 4365 in São Paulo (422.1%), and from 1257 to 3523 in Rio de Janeiro (180.3%), when comparing the years 2008 and 2022 (Table 1).

The total absolute number of deaths from CS in Brazil’s Southeast Region in children under one year of age between 2008 and 2022 was 1035, with an increase from 15 deaths in 2008 to 63 in 2022 (320.0%). The highest values were recorded in 2015 for the state of Rio de Janeiro and in 2018 for the state of São Paulo. In the comparison between the years 2008 and 2022, Minas Gerais showed an increase from 1 to 15 deaths (1400.0%); São Paulo, from 2 to 19 deaths (850.0%); and Rio de Janeiro, from 10 to 29 deaths (190.0%). However, the state of Espírito Santo showed a reduction from 2 deaths in 2008 to no deaths in 2022 (Table 1).

The incidence rate due to CS in Brazil’s Southeast Region in children under one year of age increased from 211.6 in 2008 to 1089.0 new cases per 100,000 live births in 2022 (414.6%), with higher rates in 2021 and 2022. In the comparison between the years 2008 and 2022, there was an increase from 74.7 to 944.0 in Minas Gerais (1163.7%), from 138.9 to 851.7 in São Paulo (513.2%), from 200.6 to 1086.4 in Espírito Santo (441.6%), and from 582.4 to 1953.2 new cases per 100,000 live births in Rio de Janeiro (235.4%) (Table 2).

The mortality rate due to CS in Brazil’s Southeast Region in children under one year of age increased from 1.3 in 2008 to 6.4 deaths per 100,000 live births in 2022 (392.3%), with the highest rates in 2015, 2017, and 2018. In the comparison between the years 2008 and 2022, there was an increase from 0.4 to 6.4 in the state of Minas Gerais (1500.0%), from 0.3 to 3.7 in São Paulo (1133.3%), and from 4.6 to 16.1 in Rio de Janeiro (250%). However, the state of Espírito Santo showed a decrease in the mortality rate from 3.9 deaths per 100,000 live births in 2008 to no deaths in 2021 and 2022 (Table 2).

The statistical analysis of CS incidence rates in Brazil’s Southeast Region in children under one year old, through the AAPC, revealed an increasing temporal trend of 12.8% between 2008 and 2022 (*p* < 0.001); however, a joinpoint was observed in 2016 where, despite the temporal trend remaining increasing, there was a reduction in the APC from 19.9% (*p* < 0.001) between 2008 and 2016 to 4.0% (*p* = 0.022) between 2016 and 2022 (Figure 2). In the evaluation of each state by AAPC, in the period from 2008 to 2022, an increasing temporal trend of 21.4% was observed in Minas Gerais (*p* < 0.001), 14.1% in São Paulo (*p* < 0.001), 14.0% in Espírito Santo (*p* < 0.001), and 10.9% in Rio de Janeiro (*p* < 0.001). Two joinpoints were observed in Espírito Santo and one joinpoint in Minas Gerais, Rio de Janeiro, and São Paulo. In the last time segment after the joinpoint, a stationary temporal trend was evidenced in Espírito Santo after 2020 (*p* = 0.214) and Minas Gerais after 2017 (*p* = 0.581), and an increasing temporal trend of 7.6% in Rio de Janeiro after 2012 (*p* < 0.001) and 4.6% in São Paulo after 2015 (*p* = 0.008) (Figure 2; Appendix A).

The statistical analysis of CS mortality rates in Brazil’s Southeast Region in children under one year of age through the AAPC revealed an increasing temporal trend of 11.9% between 2008 and 2022 (*p* = 0.026). Two joinpoints were observed; however, in the last temporal segment, between 2017 and 2022, the temporal trend was stationary (*p* = 0.352) (Figure 3). In the evaluation of each state, in the period from 2008 to 2022, an increasing temporal trend of 21.9% was observed in Minas Gerais (*p* < 0.001), 20.8% in São Paulo (*p* = 0.005), and 10.1% in Rio de Janeiro (*p* = 0.006). A joinpoint was observed in Rio de Janeiro in 2012, from which the temporal trend was stationary (*p* = 0.938). It was impossible to assess the temporal mortality trend in Espírito Santo because there were no deaths in 2021 and 2022 (Figure 3; Appendix A).

## 4. Discussion

Over the 15 years of the time series, 112,481 new cases of CS were recorded in children under one year of age in Brazil’s Southeast Region. There was a 346.0% increase in new cases when comparing 2008 and 2022. The incidence rate per 100,000 live births increased by 414.6% in Brazil’s Southeast Region during the period, with the highest growth in Minas Gerais (1163.7%), followed by São Paulo (513.2%), Espírito Santo (441.6%), and Rio de Janeiro (235.4%).

In 2022, the incidence rates in these states of Brazil ranged from 851.7 to 1953.2 cases per 100,000 live births, considerably higher than the global rate of 425 cases per 100,000 live births in 2020 [9], the European Union rate of 2.7 cases per 100,000 live births in 2023 [13], the USA rate of 77.6 cases per 100,000 live births in 2021 [16], the Brazilian rate of 876 per 100,000 live births in 2017 [18], and the target recommended by the WHO of 50 cases per 100,000 live births [7]. These data show that CS remains a relevant health problem in Brazil’s Southeast Region, highlighting the priority need to reevaluate specific actions to control gestational syphilis and, consequently, CS.

The temporal trend of CS incidence in Brazil’s Southeast Region increased by 12.8% between 2008 and 2022. Nevertheless, this growth trend slowed after 2016, dropping from 19.9% to 4.0%. The states of Minas Gerais, São Paulo, Espírito Santo, and Rio de Janeiro experienced increasing temporal trends of 21.4%, 14.1%, 14.0%, and 10.9%, respectively, in the same period. Despite the decline in the increasing incidence temporal trend after 2016, the absence of a decreasing temporal trend throughout the entire time series indicates that CS, although preventable, remains a substantial public health challenge in the region. This situation reflects the ongoing difficulties faced by health systems and underscores the importance of continuous improvements in prenatal care to ensure early diagnosis and treatment for pregnant women and their partners [25].

Time-series studies of CS in Brazil and the Southeast Region are limited. However, research indicates increased incidence rates of CS in Brazil in recent decades [18,19]. A study conducted in the state of Espírito Santo (2010–2019) recorded an increase from 300 to 760 cases [27], while another study in Minas Gerais (2009–2018) revealed an increase from 80 to 930 cases per 100,000 live births [28]. In the state of Rio de Janeiro, Crespo et al. (2011–2020) reported an increase from 980 to 2166 cases [29], and in São Paulo, Medeiros et al. (2007–2018) observed a rise from 130 to 660 cases per 100,000 live births [30]. The present study confirms this increase in incidence rates in all states of Brazil’s Southeast region and demonstrates persistently high values for the year 2022, reiterating that the impact of CS remains significant and reinforcing the need for collaborative efforts among healthcare professionals and health policies to effectively mitigate this serious health problem.

CS is considered an indicator of the quality of prenatal care since it is a preventable disease, and the elimination of it depends on adequate prenatal care. This includes diagnostic investigation for syphilis and cost-effective treatment with penicillin, both for the infected pregnant woman and her sexual partner [31]. The increase in the incidence rates of CS observed in Brazil’s Southeast Region reflects the increase in cases of gestational syphilis and denotes weaknesses in the public and private health systems. Although CS is a preventable disease, the efforts made to date, such as compulsory notification, government action strategies, and the “Syphilis No” Project, have not been sufficient for effective control of the disease. Investments in primary healthcare, including professional training; provision of rapid and confirmatory tests for pregnant women, sexual partners, and newborns; and access to therapy in health units are crucial for the diagnosis, treatment, and reduction of reinfections, aiming at the control of gestational syphilis and CS [7,9,25].

Between 2008 and 2022, there was a 320.0% increase in the absolute number of deaths from CS in children under one year of age in Brazil’s Southeast Region. The mortality rate per 100,000 live births increased by 392.3% in the region with significant differences, notable in Minas Gerais, where there was a growth of 1500.0%, followed by São Paulo (1133.3%) and Rio de Janeiro (250.0%). The mortality temporal trend due to CS in Brazil’s Southeast Region increased by 11.9% between 2008 and 2022. However, in the last time segment, after the joinpoint occurred in 2017, this temporal trend became stationary. The states of Minas Gerais, São Paulo, and Rio de Janeiro registered an increasing temporal trend of 21.9%, 20.8%, and 10.1%, respectively, throughout the time series. In contrast, the state of Espírito Santo showed a reduction in the mortality rate, with no records of deaths in 2021 and 2022, thus making it impossible to analyze the temporal trend in the state.

CS is the second most common infectious cause of stillbirth worldwide [7]. Data from the Centers for Disease Control and Prevention (CDC) revealed 3761 new cases of CS in 2022 in the USA, including 231 stillbirths and 51 infant deaths [32]. In the same year, in the Southeast Region of Brazil alone, 63 deaths in children under one year of age were reported, demonstrating the high number of deaths in the region. Problems such as low maternal education, unfavorable socioeconomic conditions, unprotected sexual practices, limited access to prenatal care, insufficient pediatric care in the neonatal period, and problems in the availability of penicillin for the treatment of pregnant women and newborns are factors that hinder the reduction of incidence and mortality rates due to CS [22,33].

Barragan et al., when analyzing the mortality temporal trend due to CS in the USA, pointed to an annual growth of 28.28% between 2015 and 2020 [33]. Similar results were found in Brazil’s Southeast region, with an increasing mortality temporal trend of 11.9% between 2008 and 2022, corroborating the findings. The data identified in this Brazilian study may suggest that difficulties in accessing a coordinated and qualified health system, limited resources for investment in primary care, lack of awareness about the consequences of syphilis in pregnant women and newborns, and deficiencies in monitoring and pediatric treatment of newborns with CS compromise disease control and perpetuate the risks of adverse events during pregnancy, resulting in high neonatal morbidity and mortality of affected babies.

Maternal–fetal transmission of syphilis can occur at any stage of pregnancy; although, the longer the fetal exposure to T. pallidum, the greater the risk of symptomatic infection and death of the affected newborn, especially in late-pregnancy diagnoses [2,34]. After the advent of penicillin, the elimination of CS was expected worldwide; however, what is observed in Brazil’s Southeast Region is a current and worrying temporal trend, with a significant increase in mortality rates, except in Espírito Santo. This scenario exposes the challenges in controlling syphilis and highlights the need for collective actions to reevaluate health policies for controlling gestational syphilis, as well as the urgency for diagnosis and treatment of infected newborns while still in the maternity hospital. This is extremely important to avoid late manifestations of untreated CS, which can result in irreversible sequelae with dental alterations, musculoskeletal and ophthalmological problems, neurosensory deafness, and neurodevelopmental disorders [2,3,6].

This study has some limitations. First, although CS is subject to mandatory reporting in Brazil, cases may be underreported. Second, the use of secondary databases is subject to errors in cataloging state data, in addition to the fact that the analysis considers the population group as a whole, which requires caution when transposing the results to the individual level or specific groups. Third, considering that most newborns with CS are asymptomatic at birth, this may lead to the absence of neonatal diagnosis and, consequently, a lack of adequate registration.

Finally, it is essential to acknowledge that this study did not evaluate the impact of the COVID-19 pandemic on congenital syphilis (CS). This represents a significant limitation for the last three years of the time series, as the pandemic may have affected multiple dimensions of global health, particularly regarding infectious diseases, due to social-distancing measures, lockdowns, and widespread use of face masks [35,36]. The overcrowding of primary, hospital, and emergency healthcare services may have hindered access to quality prenatal care and hospital-based maternity services, potentially compromising timely diagnosis and, consequently, influencing the incidence and mortality rates of congenital syphilis between 2020 and 2022 [35,36,37].

## 5. Conclusions

The temporal trend of CS incidence in children under one year of age increased in all of Brazil’s Southeast region states between 2008 and 2022. Incidence rates per 100,000 live births increased throughout the time series, remaining above the Brazilian average, global estimates, and those recommended by the WHO, highlighting the need to reevaluate CS control measures. The temporal trend of mortality from CS in children under one year of age also increased in Brazil’s Southeast Region between 2008 and 2022. However, in the last time segment, after 2017, a decrease in mortality was observed, with the temporal trend proving to be stationary. Mortality rates per 100,000 inhabitants increased during the time series, except for the state of Espírito Santo. Considering that CS is preventable and treatable with low-cost measures, these findings can serve as tools to assist in the evaluation, planning, and strengthening of public health policies based on current and concrete local data to minimize the impact on Brazil’s Southeast Region and achieve the goal of eliminating CS determined by the WHO.

## Figures and Tables

**Figure 1 epidemiologia-06-00022-f001:**
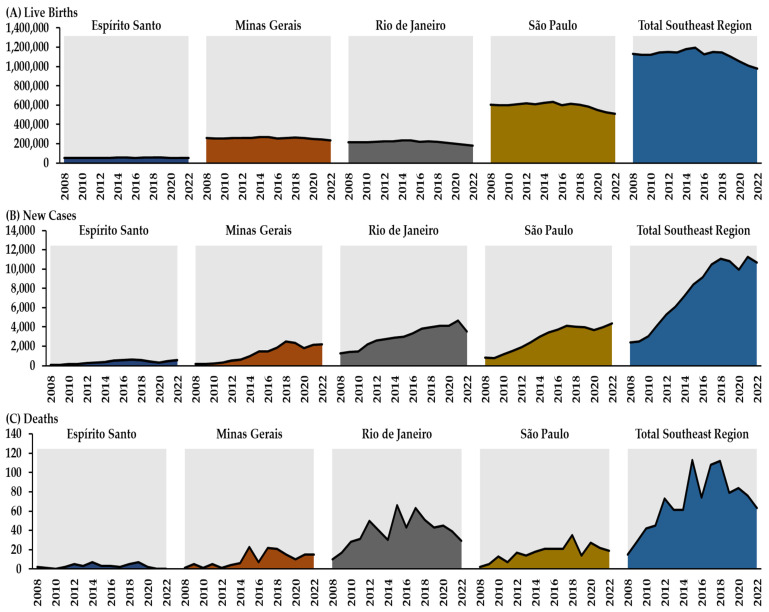
Number of live births (**A**), new cases (**B**), and deaths (**C**) due to congenital syphilis in children under one year of age in Brazil’s Southeast Region and the states of Espírito Santo, Minas Gerais, Rio de Janeiro, and São Paulo from 2008 to 2022.

**Figure 2 epidemiologia-06-00022-f002:**
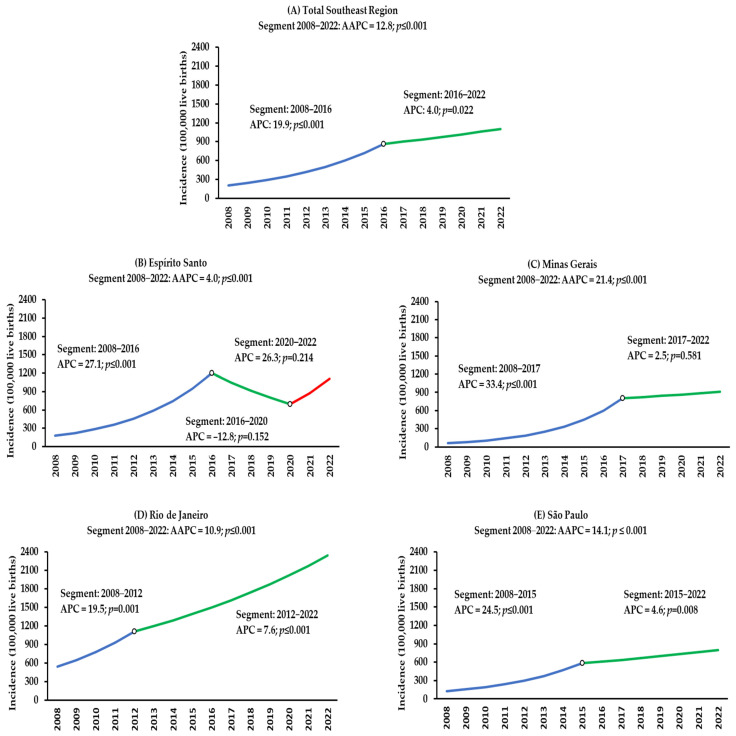
Temporal trend of congenital syphilis incidence in children under one year of age in Brazil’s Southeast Region (**A**) and the states of Espírito Santo (**B**), Minas Gerais (**C**), Rio de Janeiro (**D**), and São Paulo (**E**) from 2008 to 2022.

**Figure 3 epidemiologia-06-00022-f003:**
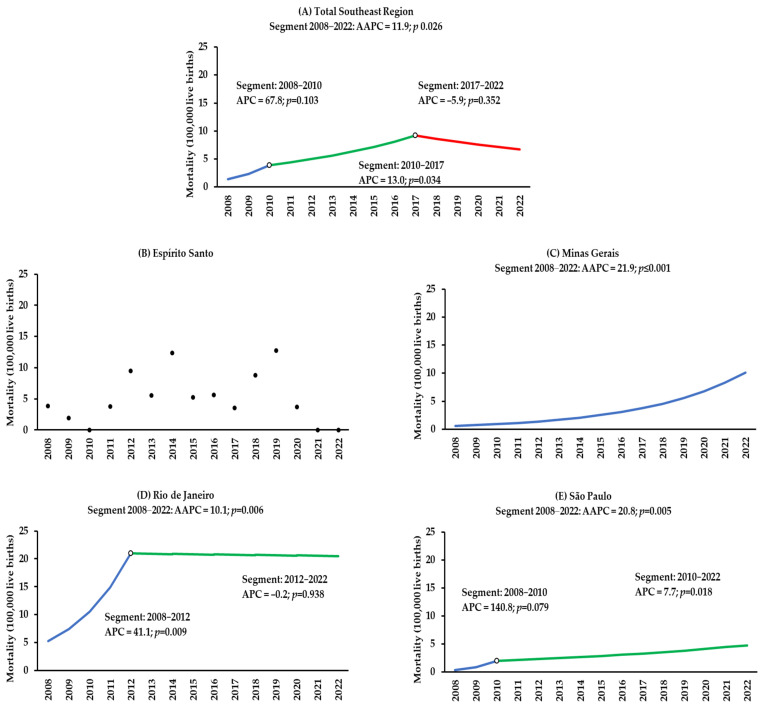
Temporal trend of congenital syphilis mortality in children under one year of age in Brazil’s Southeast Region (**A**) and the states of Espírito Santo (**B**), Minas Gerais (**C**), Rio de Janeiro (**D**), and São Paulo (**E**) from 2008 to 2022.

**Table 1 epidemiologia-06-00022-t001:** Number of new cases and deaths due to congenital syphilis in children under one year of age in Brazil’s Southeast Region and the states of Espírito Santo, Minas Gerais, Rio de Janeiro, and São Paulo from 2008 to 2022.

	Espirito Santo		Minas Gerais		Rio De Janeiro		São Paulo		Total Southeast Region	
	New Cases	Deaths	New Cases	Deaths	New Cases	Deaths	New Cases	Deaths	New Cases	Deaths
2008	104	2	195	1	1257	10	836	2	2392	15
2009	96	1	195	5	1395	17	796	5	2482	28
2010	158	0	227	1	1468	28	1167	13	3020	42
2011	170	2	313	5	2182	31	1496	7	4161	45
2012	279	5	505	1	2580	50	1923	17	5287	73
2013	338	3	645	4	2734	40	2390	14	6107	61
2014	382	7	948	6	2884	30	3007	18	7221	61
2015	519	3	1439	23	3001	66	3439	21	8398	113
2016	600	3	1487	7	3326	43	3727	21	9140	74
2017	648	2	1840	22	3848	63	4125	21	10,461	108
2018	565	5	2477	21	3998	51	4035	35	11,075	112
2019	443	7	2329	15	4105	43	3955	14	10,832	79
2020	339	2	1804	10	4125	45	3682	28	9950	85
2021	480	0	2171	15	4645	39	3990	22	11,286	76
2022	562	0	2219	15	3523	29	4365	19	10,669	63

**Table 2 epidemiologia-06-00022-t002:** Incidence and mortality rates of congenital syphilis in children under one year of age in Brazil’s Southeast Region and the states of Espírito Santo, Minas Gerais, Rio de Janeiro, and São Paulo from 2008 to 2022.

	Espirito Santo		Minas Gerais		Rio De Janeiro		São Paulo		Total Southeast Region	
	Incid	Mort	Incid	Mort	Incid	Mort	Incid	Mort	Incid	Mort
2008	200.6	3.9	74.7	0.4	582.4	4.6	138.9	0.3	211.6	1.3
2009	186.6	1.9	77.2	2.0	644.0	7.8	133.0	0.8	221.8	2.5
2010	304.7	0.0	89.0	0.4	682.0	13.0	194.1	2.2	268.8	3.7
2011	320.4	3.8	120.4	1.9	989.1	14.1	245.2	1.1	363.8	3.9
2012	528.1	9.5	193.8	0.4	1157.7	22.4	311.9	2.8	458.6	6.3
2013	625.2	5.5	249.4	1.5	1220.4	17.9	391.2	2.3	532.1	5.3
2014	675.5	12.4	354.9	2.2	1234.7	12.8	480.6	2.9	610.4	5.2
2015	911.5	5.3	536.3	8.6	1266.5	27.9	542.4	3.3	702.0	9.4
2016	1123.3	5.6	586.5	2.8	1517.8	19.6	619.7	3.5	810.6	6.6
2017	1160.3	3.6	705.1	8.4	1723.8	28.2	674.2	3.4	908.2	9.4
2018	996.1	8.8	939.5	8.0	1813.2	23.1	665.7	5.8	965.6	9.8
2019	806.6	12.7	906.6	5.8	1973.7	20.7	678.2	2.4	982.1	7.2
2020	630.5	3.7	729.8	4.0	2071.6	22.6	666.7	5.1	945.5	8.1
2021	914.4	0.0	896.6	6.2	2446.5	20.5	759.7	4.2	1117.7	7.5
2022	1086.4	0.0	944.0	6.4	1953.2	16.1	851.7	3.7	1089.0	6.4

Incidence rate per 100,000 live births. Mortality rate per 100,000 live births.

## Data Availability

Data were extracted from https://datasus.saude.gov.br/informacoes-de-saude-tabnet/ (accessed on 31 December 2024).

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
