# Peer review of "Trends in Congenital Syphilis Incidence and Mortality in Brazil’s Southeast Region: A Time-Series Analysis (2008–2022)"

_epidemiologia, 2025, doi:10.3390/epidemiologia6020022_

Round 1

Reviewer 1 Report

Comments and Suggestions for Authors

The article is particularly interesting for the topics addressed and the results obtained, which highlight how early diagnosis and appropriate screening methods for syphilis remain crucial even today. This is essential to prevent serious complications, especially during pregnancy and in childhood, as clearly described by the Authors.

I would just like to ask the Authors a few questions:

1) How do you explain the observed differences in the trends of incidence and mortality of congenital syphilis between 2008 and 2022? Could a change in the reporting methods and data acquisition by the DATASUS be among the possible explanations?

2) Are there any epidemiological differences compared to other regions of Brazil? If so, what could they be attributed to?

Author Response

Comments 1: How do you explain the observed differences in the trends of incidence and mortality of congenital syphilis between 2008 and 2022? Could a change in the reporting methods and data acquisition by the DATASUS be among the possible explanations?

Response 1: Thank you for pointing this out. In our assessment, several factors have contributed to the increase in the incidence and mortality of congenital syphilis in Brazil between 2008 and 2022. Below, we highlight the relevant aspects:

a) It is unlikely that changes in diagnostic criteria or the notification methods adopted by DATASUS would explain the study's findings. During the period analyzed, no significant modifications were identified in the congenital syphilis notification criteria, which has remained a compulsory notifiable disease in Brazil for many years. The only modification occurred in 2017 and referred to the removal of the requirement for sexual partner treatment as a criterion to define adequate maternal treatment. However, this change would not impact our findings, as the diagnostic criteria for congenital syphilis in Brazil do not take into account the partner's test results or treatment but rely exclusively on the clinical and laboratory findings of the mother and the newborn. Therefore, it is improbable that variations in the reporting systems account for the significant increase observed in incidence and mortality indicators.

b) A determining factor, in our view, is the low coverage of prenatal care, especially among women with low income and low educational attainment. Many pregnant, particularly in regions with inadequate infrastructure, do not receive appropriate prenatal care and laboratory tests for early diagnosis and adequate treatment of the infection. Additionally, the absence of adequate treatment for sexual partners facilitates reinfection during pregnancy, even in women who initially received treatment.

c) Between 2014 and 2016, Brazil faced problems in the supply of penicillin, the first-line treatment for syphilis. Although the supply was later restored, this period may have contributed to the worsening of the situation, although it does not fully explain all the findings.

d) Another important point is that many pregnant women are asymptomatic and, by not undergoing prenatal care and serological testing during pregnancy, are only diagnosed at the time of delivery, when the newborn is already infected, often with severe multisystem involvement, contributing to neonatal mortality.

e) Finally, unprotected sexual practices remain common in Brazil, combined with a lack of information and limited access to preventive methods through the public health system. These factors increase women's risk of infection, resulting in prolonged fetal exposure to Treponema pallidum, especially in cases of early gestational infection, raising the risk of congenital syphilis with severe outcomes and sequelae when not treated adequately and promptly.

f) Considering all these aspects, it is notable that, despite the efforts and control strategies implemented by the Ministry of Health, Brazil continues to report high levels of incidence and mortality associated with a condition that is both preventable and treatable at low cost.

g) It is also important to underline that the mortality data presented in the study refer exclusively to deaths among live births, excluding miscarriages and stillbirths, which suggests that the actual mortality rates may be even higher. Regarding incidence, we also believe it may be underestimated, as many newborns are asymptomatic and, if they do not undergo neonatal screening, may not be reported initially.

Comments 2: Are there any epidemiological differences compared to other regions of Brazil? If so, what could they be attributed to?

Response 2: Thank you for the comments. Few studies from other regions of Brazil employ a methodology similar to that used in the present study, especially regarding mortality. However, we selected two relevant studies that describe differences among the regions of Brazil (references 18 and 19).

Study 18 observed a growing trend in the incidence of congenital syphilis between 2007 and 2017 across all Brazilian regions, with the worst indicators reported in the South region, followed by the Southeast and Northeast. Study 19 identified increased incidence rates in all five Brazilian regions from 2008 to 2018, with the highest rates in the Southeast, followed by the Northeast and South.

These two studies reveal variation in incidence among the regions; however, both demonstrate a consistent upward trend over the time series analyzed.

Despite congenital syphilis being a preventable and treatable condition through early diagnosis and intervention, the reasons for these increasing trends are similar across the different studies. Key factors highlighted include limited access to and low-quality prenatal care, low rates of adequate treatment among pregnant women, the need for more significant investment in public health policies, the lack of large-scale availability of diagnostic tests, difficulties in accessing appropriate treatment within primary care settings for women with syphilis and their sexual partners and considering Brazil's socioeconomic context, which is marked by poverty and low educational attainment, especially in regions with poor infrastructure, which undermines effective health education, diagnosis, treatment, and case follow-up.

We reviewed additional epidemiological studies conducted in states within the Southeast region (references 27, 28, 29, and 30), whose findings align with those presented in this manuscript. We also identified a study conducted in the state of Pará, in the Northern region (Brazilian Amazon), from 2007 to 2020, and another in the state of Ceará, in the Northeast region, from 2009 to 2018. Both studies reported an increase in congenital syphilis incidence rates. In Ceará, a higher number of cases was found among pregnant women aged 15 to 29 years with incomplete primary education. Moreover, approximately 39% of the pregnant women were diagnosed only at the time of delivery. Even among those diagnosed during pregnancy, many did not receive treatment or were inadequately treated, highlighting the weakness of prenatal care in these settings.

Reviewer 2 Report

Comments and Suggestions for Authors

The paper addresses the important topic of global health because of growing trends in CS that also affect countries with more favourable indicators. I see that significant changes have already been made. I suggest some additions and minor editorial changes.

The authors refer to the ECDC report on adults. A reference to a more recent similar report on CS was definitely missing. This is a new publication, probably not available at the time of submitting this paper (European Centre for Disease Prevention and Control. Congenital syphilis. In: ECDC. Annual epidemiological report for 2023. Stockholm: ECDC; 2025.). In particular, it is worth referring to the global target cited on page 6 of this report and the distance between European and South American indicators. There is an appropriate reference in the text at line 59, but incomplete. Is it not about the indicated decline in 80% of countries.

In addition, the authors discuss the sources of the data (lines 117+). It is worth noting here the extent to which detected cases are confirmed by laboratory tests. Have criteria changed during the period studied that may have influenced better detection? The weaknesses of the data collection system are mentioned in the limitations of the study, but it would have been worth describing this system better at the outset, also comparing it with European standards.

A key issue related to CS is the quality of the prenatal care system. As complements, the authors have added a paragraph about Brazil's inclusion in the EMTCT initiative. The reader can expect a description of the standards of care for pregnant women in the context of CS prevention. Has anything changed during the period under study or are other standards currently being implemented in relation to the above initiative.

The trends described in the conclusions are inconsistent. In the following sentences, there is mention of a decrease and an increase. In addition, mortality per 100,000 inhabitants appears, which has not been presented before (with the denominator so specified). If anywhere there is data for the whole population, this needs to be highlighted and the denominator (line 97+?) indicated. I find it difficult to review the whole text under this aspect.

As far as editorial comments are concerned, I would suggest that in the tables and figures next to Southeast Region you write Total SR. Without knowing the specifics of the country, it can be assumed that this is not the total result but the fifth region. Southeast Region is interchangeably written as Southeast region.

There is no description of the vertical axis in part (a) of Figure 2. 

Why was section 6.Patents added, which does not apply to this paper.

Author Response

Comments 1: The authors refer to the ECDC report on adults. A reference to a more recent similar report on CS was definitely missing. This is a new publication, probably not available at the time of submitting this paper (European Centre for Disease Prevention and Control. Congenital syphilis. In: ECDC. Annual epidemiological report for 2023. Stockholm: ECDC; 2025.). In particular, it is worth referring to the global target cited on page 6 of this report and the distance between European and South American indicators. There is an appropriate reference in the text at line 59, but incomplete. Is it not about the indicated decline in 80% of countries.

Response 1: Thank you for pointing this out and for the new ECDC reference. Indeed, this publication was not yet available when we wrote the manuscript. The necessary adjustments have now been made to include the updated congenital syphilis case rates for the European Union for the years 2022 and 2023.

Line 53 mentions the global target, which is described as “in 80% of countries.” The specific target for the European Region has also been included in lines 56, 57, and 58.

Reference 13 has been updated accordingly in lines 471, 472, and 473.

Comments 2: In addition, the authors discuss the sources of the data (lines 117+). It is worth noting here the extent to which detected cases are confirmed by laboratory tests. Have criteria changed during the period studied that may have influenced better detection? The weaknesses of the data collection system are mentioned in the limitations of the study, but it would have been worth describing this system better at the outset, also comparing it with European standards.

Response 2: Thank you for the comments. According to the criteria established by the Brazilian Ministry of Health, children are considered to have a diagnosis of congenital syphilis when they meet specific laboratory criteria, particularly a positive non-treponemal test. This test is present in all cases of congenital syphilis, except when the mother, despite being diagnosed with syphilis, was either untreated or inadequately treated or when there is microbiological evidence of Treponema pallidum infection in samples from secretions, skin lesions, biopsies, or autopsies.

Criteria used by the Brazilian Ministry of Health for the diagnosis of congenital syphilis:

Scenario 1: Any live birth, stillbirth, or miscarriage from a woman with untreated or inadequately treated syphilis.

Scenario 2: Any child under 13 years of age presenting with at least one of the following conditions:

  1. a) Clinical manifestations, cerebrospinal fluid, or radiologic alterations consistent with congenital syphilis and a reactive non-treponemal test;
  2. b) Non-treponemal test titers in the infant at least two dilutions higher than those of the mother, based on peripheral blood samples collected simultaneously at birth;
  3. c) A fourfold increase in non-treponemal test titers during follow-up of an exposed infant;
  4. d) Persistently reactive non-treponemal tests after six months of age in a child adequately treated during the neonatal period;
  5. e) Reactive treponemal tests after 18 months of age without a prior diagnosis of congenital syphilis.

Scenario 3: Microbiological evidence of Treponema pallidum infection from a sample of nasal discharge or skin lesion, biopsy, or autopsy of a child, miscarriage, or stillbirth.

Regarding the second question, no significant changes were identified during the analyzed period in the congenital syphilis case definition criteria, which has remained a notifiable disease in Brazil for many years. The only modification occurred in 2017 and involved the removal of the requirement for sexual partner treatment as a condition to define adequate maternal treatment. However, this change does not impact our findings, as the diagnostic criteria for congenital syphilis in Brazil are based on clinical and laboratory findings from the mother and the newborn, without considering the partner’s test results or treatment.

Comments 3: A key issue related to CS is the quality of the prenatal care system. As complements, the authors have added a paragraph about Brazil's inclusion in the EMTCT initiative. The reader can expect a description of the standards of care for pregnant women in the context of CS prevention. Has anything changed during the period under study or are other standards currently being implemented in relation to the above initiative.

Response 3: We agree with this comment. Lines 80–88 describe the modifications and strategies the Brazilian Ministry of Health implemented to meet the WHO target.

This article also aims, secondarily, to assess the effectiveness of these measures in a key region of Brazil.

Comments 4: The trends described in the conclusions are inconsistent. In the following sentences, there is mention of a decrease and an increase. In addition, mortality per 100,000 inhabitants appears, which has not been presented before (with the denominator so specified). If anywhere there is data for the whole population, this needs to be highlighted and the denominator (line 97+?) indicated. I find it difficult to review the whole text under this aspect.

Response 4: Our study revealed an overall increasing trend in the incidence of congenital syphilis between 2008 and 2022, although a deceleration was observed after 2016 (from 19.9% to 4.0%). Despite this slowdown, incidence and mortality rates remain high in the region, highlighting how far we are from achieving the WHO target and underscoring the urgent need to improve strategies for the elimination of congenital syphilis.

Lines 272–273 were removed to improve clarity in the presentation of our findings.

The mortality rate per 100,000 live births is described in the Methods section, the Results (lines 196–202), and the Conclusion. The mortality temporal trend analysis revealed an overall increasing trend over the time series, with a joinpoint identified in 2017, indicating a shift to a stationary trend (p = 0.352).

Still, in response to Comments 4, all data were calculated based on the number of live births in each state and across the entire region.

Comments 5: As far as editorial comments are concerned, I would suggest that in the tables and figures next to Southeast Region you write Total SR. Without knowing the specifics of the country, it can be assumed that this is not the total result but the fifth region. Southeast Region is interchangeably written as Southeast region.

Response 5: We agree. We apologize if the description of the Southeast Region of Brazil was unclear. This region comprises the four states analyzed individually and collectively in the present study: São Paulo, Rio de Janeiro, Espírito Santo, and Minas Gerais. We standardized the term as “Southeast Region” with capital letters rather than “Southeast region.”

We have organized all of the Total Southeast Region in the tables.

Comments 6: There is no description of the vertical axis in part (a) of Figure 2. 

Response 6: We agree. We apologize for the omission of the vertical axis label in Figure 2. The correction has been made.

Comments 7:  Why was section 6. Patents added, which does not apply to this paper.

Response 7: We apologize for the mistakes. The adjustment has been made as requested.